# Tumour sampling conditions perturb the metabolic landscape of clear cell renal cell carcinoma

Cissy Yong[1,2,3], Christina Schmidt [4], Ming Yang[4], Alexander Von Kriegsheim [5], Anne Y. Warren [6], Shubha Anand[7], James N. Armitage[1], Antony C. P. Riddick[1], Thomas J. Mitchell[1,2,3], Vishal Patil [8], Kourosh Saeb-Parsy [2], Sakari Vanharanta [9], Grant D. Stewart[1,2,3,11] & Christian Frezza [4,10,11] ✉

Human isotopic tracer studies are key for in vivo studies of cancer metabolism. Yet, the effects of sampling conditions on the tissue metabolome remain understudied. Here, we perform a $^{13}C$-glucose study coupled with metabolomic, transcriptomic, and proteomic profiling in patients with clear cell renal cell carcinoma (ccRCC) to assess the impact of ischaemia on tissues sampled intraoperatively and post-surgical resection, where tissues are exposed to varying degrees of warm ischaemia. Although several metabolic features were preserved, including suppressed TCA cycle activity, ischaemia masked other metabolic phenotypes of ccRCC, such as suppressed gluconeogenesis. Notably, normal kidneys were more metabolically susceptible to ischaemia than the ccRCC tumours. Despite their overall stability, ischaemia caused subtle changes in the proteome and transcriptome. Using orthotopic ccRCC-derived xenografts, we evidenced that prolonged ischaemia disrupted the tissue metabolome stability. Overall, minimising tissue ischaemia is pivotal in accurately profiling cancer metabolism in patient studies.

Renal cell carcinoma (RCC) is the most lethal urological malignancy with a 10-year survival rate of 52%[1,2]. RCC is established as a metabolically-driven disease in which metabolic reprogramming is implicated in renal tumourigenesis, progression, and aggressive phenotypes[3–8]. Consequently, great strides have been made to identify these metabolic perturbations for translation into clinical practice. One such achievement is the development of isotopic (e.g. $^{13}Carbon$, $^{15}Nitrogen$, $^{2}Hydrogen$) nutrient tracers that can be introduced into a biological system to identify the metabolic fates within tissues using

mass spectrometry[9]. Over the last decade, these models have been increasingly adapted to study cancer metabolism in vivo in patients undergoing surgical resection of tumours, including in lung[10,11], brain[12], paediatric solid tumours[13], and in clear cell RCC (ccRCC)[14]. The operating theatre setting permits a controlled environment for intravenous tracer administration to patients in conjunction with rapid acquisition of tissues for analysis. However, the principles of surgical oncology typically involves devascularisation of the organ and/or tumour early on in the procedure to enable safe removal, which subjects the tissues

[1]Department of Urology, Addenbrooke's Hospital, Cambridge University Hospitals NHS Foundation Trust (CUHFT), Cambridge, UK. [2]Department of Surgery, University of Cambridge, Cambridge, UK. [3]Urological Malignancies Virtual Institute, CRUK Cambridge Centre, University of Cambridge, Cambridge, UK. [4]University of Cologne, Faculty of Medicine and University Hospital Cologne, Institute for Metabolomics in Ageing, Cologne, Germany. [5]Edinburgh Cancer Research UK Centre, Institute of Genetics and Molecular Medicine, Edinburgh, UK. [6]Department of Pathology, Addenbrooke's Hospital, CUHFT, Cambridge, UK. [7]Cancer Molecular Diagnostics Laboratory, CRUK Cambridge Cancer Centre, Cambridge, UK. [8]Department of Anaesthetics, Addenbrooke's Hospital, CUHFT, Cambridge, UK. [9]Translational Cancer Medicine Program, Faculty of Medicine, University of Helsinki, Helsinki, Finland. [10]Faculty of Mathematics and Natural Sciences, Institute of Genetics, Cluster of Excellence Cellular Stress Responses in Aging-associated Diseases (CECAD), Cologne, Germany. [11]These authors jointly supervised this work: Grant D. Stewart, Christian Frezza. ✉e-mail: christian.frezza@uni-koeln.de

to a duration of warm ischaemia before conventional research tissue sampling can occur. Whilst a lack of studies have assessed the effects of tissue ischaemia on the tumour metabolome, ischaemia-reperfusion injury studies in transplantation research have identified significant metabolic alterations in benign mammalian tissues exposed to ischaemia[15–17]. Given that the metabolome forms the primary evidence to infer cancer-driven metabolic reprogramming, it is critical to elucidate the impact of ischaemia, as a consequence of surgical tissue sampling, on human tumour phenotypes. Similarly, the impact on other more thoroughly investigated molecular layers, such as the proteome and transcriptome, is unclear. As human tissue analyses become increasingly accessible through the growing number of collaborative human biorepositories, it is imperative to evaluate this technical factor to ensure accurate molecular profiling. Here, we apply a multiomics approach to a $^{13}C_6$-glucose study in patients with ccRCC undergoing surgery with tissues analysis at two timepoints, intraoperatively or 'in vivo' (tissues perfused) and post-surgical resection (tissues ischaemic). We demonstrate suppression of gluconeogenesis in these ccRCC tumours in vivo and the significant impact of ischaemia on the metabolic characterisation of these tumours. Furthermore, using a xenograft mouse model, we show that prolonged exposure of tissues to ischaemia is the key determinant to this metabolic perturbation we observe. Collectively, this study has critical implications on future human tissue sampling methods, specifically in minimising tissue exposure to ischaemia, in order to accurately profile the tumour metabolome.

## Results

### Metabolic dysregulation in ccRCC tumours in vivo

To investigate the metabolism of ccRCC with an intact blood supply, we performed metabolomic, transcriptomic, and proteomic analyses on intraoperative multiregional samples taken from 5 patients with *VHL*-mutant ccRCC undergoing $^{13}C_6$-glucose infusions at the time of radical nephrectomy surgery (Fig. 1A and Table 1). Key annotation information about the patients is provided in Supplementary Table 1. Importantly, no perioperative complications arose from sampling tissues intraoperatively (tumour and normal adjacent kidney). Supervised analysis of the metabolomics data revealed tissue-specific clustering with more prominent dispersion within the tumour cluster, indicating metabolic heterogeneity (both intra- and inter-heterogeneity) in our ccRCC cohort, as previously reported[18–21] (Fig. 1B, 1C). Several metabolic features that were shown to differentially accumulate in previous snapshot (unlabelled) metabolomics studies in ccRCC were also identified in our cohort (Supplementary Fig. 1A), including 2-hydroxyglutarate[5,8] and tryptophan catabolites[22–24]. Interestingly, applying hierarchical clustering revealed two main metabolic clusters in our tumour samples with differentially expressed features detected from two metabolic pathways: urea cycle and tryptophan pathway within these clusters (Fig. 1C and Supplementary Fig. 1B). Analysis of the cluster-specific patterns within these pathways revealed clusters that were associated with the fold change amplitude e.g. ornithine and kynurenic acid, or with fold change divergence e.g. argininosuccinate (Fig. 1D and Supplementary Fig. 1C). However, the corresponding urea cycle enzymes, argininosuccinate synthase-1 (ASS1) and argininosuccinate lyase (ASL), which are involved in argininosuccinate production and consumption respectively, were overall suppressed in these tissues and did not correlate with the cluster-specific dichotomous metabolite levels observed (Fig. 1D and Supplementary Fig. 1D).

Molecular characterisation of these tumours revealed a strong protein-RNA correlation (Rho =0.76) with gene set enrichment analysis (GSEA) corroborating the characteristic hallmarks of ccRCC (e.g. upregulated glycolysis, hypoxia, angiogenesis, and epithelial mesenchymal transition) at both the protein and RNA level (Rho=0.85) (Fig. 1E and Supplementary Fig. 1E). Focussed GSEA using an established metabolic gene signature set[25] corroborated several

dysregulated metabolic pathways reported in ccRCC including OXPHOS, tryptophan, and the urea cycle pathways. Strikingly, an overall downregulation of 'glycolysis and gluconeogenesis' proteins was observed in our cohort despite the well-characterised upregulated glycolytic activity in ccRCC tumours[4,6,14,26]. This result suggested that RNA or protein expression is not sufficient to dissect the metabolic activity in ccRCC tumours, and an orthogonal approach is required (Fig. 1F).

### Gluconeogenesis is suppressed in ccRCC tumours in vivo

To comprehensively probe central carbon metabolism activity in ccRCC tumours with an intact blood supply, we performed isotopic $^{13}C$ tracing in patients (Supplementary Fig. 2A). Serum enrichment of $^{13}C_6$-glucose (29-31%) in our cohort was comparable to previous studies[14] (Supplementary Fig. 2B). Higher glycolytic labelling from $^{13}C_6$-glucose (hexose phosphates, HP m + 6 and lactate m + 3), elevated total lactate pools, and significantly reduced labelling of several TCA cycle intermediates (e.g. citrate, succinate, fumarate) was observed (Fig. 2A, B and Supplementary Fig. 2C), corroborating the characterised aerobic glycolysis in these tumours[6,14]. This is corroborated by the significantly elevated enzyme levels of LDHA, which catalyses the final step of glycolysis (Fig. 2C). Of note, these findings are consistent with the high LDHA/low LDHB expression in ccRCC. However, total metabolite pools of TCA cycle intermediates were either comparable between tissues or significantly elevated (α-ketoglutarate, a-KG; and glutamine) in these tumours (Fig. 2B). These results suggest the presence of a large unlabelled (m + 0) pool in tumour tissue, which alongside the accumulated levels of glutamine and a-KG (the TCA cycle entry metabolite for glutamine anaplerosis), indicated glutamine as an anaplerotic source in ccRCC as previously proposed[14,26–28] (Fig. 2B and Supplementary Fig. 2C).

Strikingly, despite increased labelling, the total pools of glycolytic intermediates (e.g. HP; 2/3-phosphoglycerate, 2/3-PG) were overall lower in ccRCC tumours compared with normal kidney tissues (Fig. 2B). Furthermore, a dichotomous labelling pattern in HP showed significantly lower levels of HP m + 1 and m + 3 in these tumours (Fig. 2A and Supplementary Fig. 2D). This metabolite labelling profile primarily arises from bidirectional metabolic activity (Supplementary Fig. 2A). However, given the negligible circulating enrichment of labelled glucose isotopologues (aside from the infused glucose m + 6) (Supplementary Fig. 2E), this pattern suggested suppressed gluconeogenic activity in ccRCC compared with kidneys. This is further corroborated by the negligible labelling of glucose m + 3 or m + 1 within both tissues, which we would have observed if the HP labelling patterns were derived from glycolytic activity (Fig. 2A, Supplementary Fig. 2C). Indeed, our proteomics analysis revealed significant depletion of gluconeogenic-specific enzymes such as fructose-1,6-bisphosphataste 1 (FBP1) and phosphoenolpyruvate carboxykinase (PCK), which was strongly correlated (Rho=0.87) at the transcriptomic level (Fig. 2C and Supplementary Fig. 2F). Altogether, our multiomics approach complemented with $^{13}C_6$-glucose labelling provides evidence for the suppression of gluconeogenic activity in human ccRCC tumours in vivo (Fig. 2D).

### Post-operative ischaemia perturbs the metabolic landscape of ccRCC tumours

To assess the impact of warm ischaemia time (WIT) on the molecular characterisation of human tumours, we performed metabolomic, proteomic, and transcriptomic analyses on paired tissue samples (ccRCC and normal kidney) taken post-operatively (Fig. 3A). A variable WIT (10-60 mins) was observed, which was calculated from the time of renal artery ligation intraoperatively to the earliest feasible time point where research tissue sampling occurred (Supplementary Table 1). Unsupervised cluster analysis revealed stark contrasts between our molecular datasets (Fig. 3B). No clustering was observed based on

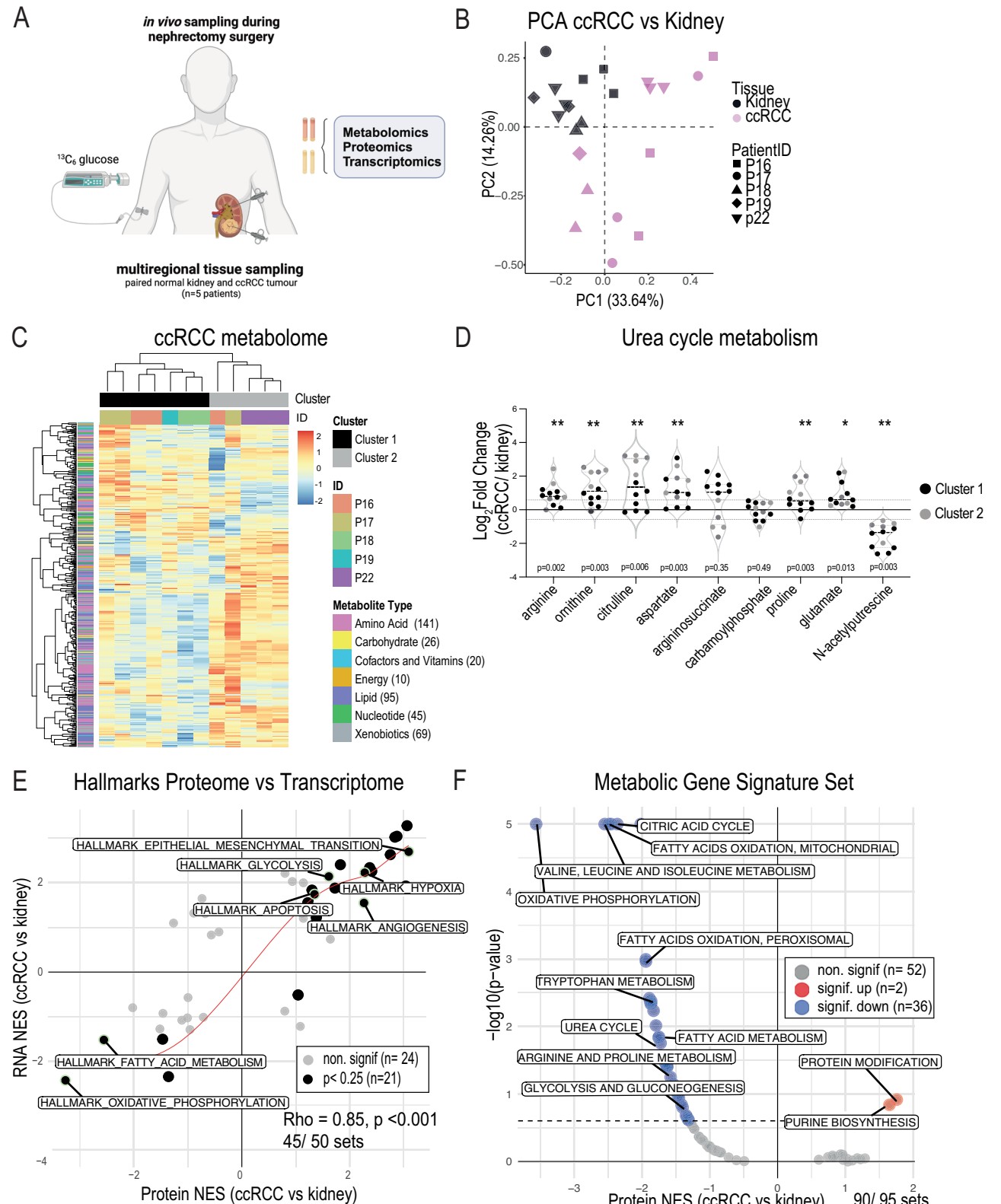

**A** *in vivo* sampling during nephrectomy surgery

Metabolomics
Proteomics
Transcriptomics

$^{13}C_6$ glucose

**multiregional tissue sampling**
paired normal kidney and ccRCC tumour
(n=5 patients)

**B** PCA ccRCC vs Kidney

**C** ccRCC metabolome

**D** Urea cycle metabolism

**E** Hallmarks Proteome vs Transcriptome

**F** Metabolic Gene Signature Set

tissue type (kidney vs ccRCC) or condition (WIT vs in vivo) in the metabolome, whereas the proteome and transcriptome demonstrated tissue-specific clustering irrespective of exposure to WIT. Furthermore, comparison of the significantly differentiated features within each dataset revealed the metabolome to have the largest fold change amplitude differences between ischaemia and in vivo conditions, relative to the proteome and transcriptome (Supplementary Fig. 3A).

The metabolic GSEA of the proteome was also strongly correlated (Rho=1) between conditions, indicating general stability at the protein level (Supplementary Fig. 3B).

Notably, central carbon metabolism and nucleotide (purine and pyrimidine) degradation pathway metabolites were differentially accumulated in ischaemic compared to control (in vivo) kidneys, and to a lesser extent in ccRCC tumours (Fig. 3C, Supplementary Fig. 3C).

**Fig. 1 | Multimodal characterisation of clear cell renal cell carcinoma (ccRCC) tumours in vivo.** See also Supplementary Fig. 1. **A** Schematic of in vivo patient tissue sampling during surgery (intraoperative) and the planned multiomics analyses. **B** Principal Component Analysis (PCA) plot on metabolomics data from ccRCC tumours (pink) and kidneys (black). $n = 5$ patients (symbol shape) with 1- 3 multiregional samples per patient tissue, see Supplementary Table 1. **C** Heatmap of metabolite fold change (ccRCC vs kidney) for all matched patient (ID colour) tumour samples ($n = 12$ tumour samples taken from 5 patients). Cluster (Cluster colour) was assigned based on hierarchal clustering and metabolites are grouped by metabolite type (Metabolite type colour). **D** Violin plot of urea cycle metabolite Log2FC (ccRCC vs kidney) showing the assigned clusters from (**C**) with cluster 1 (black) and cluster 2 (grey). Violin plots display the median, 1st and 3rd quartiles.

$p = 0.002$ (arginine), $p = 0.003$ (ornithine), $p = 0.006$ (citrulline), $p = 0.003$ (aspartate), $p = 0.003$ (proline), $p = 0.013$ (glutamate), and $p = 0.003$ (N-acetylputrescine). **E** Correlation plot comparing protein and RNA normalised enrichment scores (NES) for Hallmarks Gene Set Enrichment Analysis (GSEA) (ccRCC vs kidney). Symbol colour denotes significant NES scores (black) and non-significant scores (grey). **F** Volcano plot of protein NES for metabolic gene signature sets (MGSS) GSEA (ccRCC vs kidney). Symbol colour denotes significantly up (red), significantly down (blue), and non-significant scores (grey). *$p < 0.05$, **$p < 0.01$ by two-sided Student's t-test using Benjamini-Hochberg correction (**D**). Correlation coefficients calculated using two-sided Spearman's method with LOESS regression line fitted (**E**). Source data are provided as a Source Data file. (**A**) created in BioRender. Burge, S. (2025) https://BioRender.com/n43g446.

### Table 1 | Patient demographics and labelled glucose infusions details

| ID | Age | Sex | Surgery (radical nephrectomy) | Histology | VHL status | Tumour grade[a] | Pathology Staging[b] | Infusion (mins) | Blood Glucose[c] (mmol/L) |
|----|-----|-----|-------------------------------|-----------|------------|-----------------|----------------------|-----------------|---------------------------|
| P16 | 70-80 | M | open (right) | ccRCC | Mutant | 4 | pT3a | 120 | 8.3 |
| P17 | 60-70 | M | open (left) | ccRCC | Mutant | 4 | pT4 pN1 | 145 | 8.4 |
| P18 | 70-80 | M | open (left) +renal vein thrombectomy | ccRCC | Mutant | 4 | pT3a pM1 | 120 | 7.6 |
| P19 | 60-70 | M | open (left) | ccRCC | Mutant | 3 | pT3a | 95 | 8.6 |
| P22 | 50-60 | M | open (right) | ccRCC | Mutant | 4 | pT3a | 102 | 8.8 |

*ccRCC* clear cell renal cell carcinoma. *VHL* Von Hippel Lindau gene. [a]WHO/ISUP grading system[71]. [b]TNM staging 8th Edition[72]. [c]Blood glucose concentration at time of tissue sampling intraoperatively.

Several of these metabolites (succinate, hypoxanthine, and xanthine) have been previously characterised as markers of ischaemia across multiple mammalian tissue including in human kidneys[15–17] and were similarly elevated in both tissues exposed to ischaemia (albeit to a lesser extent in ccRCC) compared to control conditions (Fig. 3D). Lactate, which is a marker of anaerobic metabolism and implicated in the ischaemic/hypoxic cellular response, was also elevated in tissues. As several central carbon pathway metabolites were differentially altered in a tissue-specific manner with ischaemia, we examined the impact this would have on determining metabolite abundance (Log2 fold change, Log2FC) in ccRCC between conditions (Fig. 3E). Whilst most metabolites maintained Log2FC directionality, suggesting that the metabolic changes in individual ischaemic tissues were grossly proportional to control tissues, several Log2FC amplitudes significantly differed between conditions. Moreover, some metabolites (HP; dihydroxyacetone phosphate, DHAP; and aconitate) displayed opposite Log2FC expressions in ischaemic compared with control conditions. These evident disparities could vastly alter phenotypic interpretation, particularly in snapshot metabolomics studies.

Alteration to the labelling profile of central carbon pathway metabolites was evident in our ischaemic versus control ccRCC tumours (Supplementary Fig. 3D). These changes were likewise observed at the tissue-specific level with several metabolites displaying a general accumulation (e.g. lactate and succinate) or depletion (e.g. HP and aconitate) of their labelled isotopologues with exposure to ischaemia (Fig. 3F). Ischaemia-induced reversal of tissue metabolite labelling levels were most appreciated in lactate and HP isotopologues (Fig. 2A and Supplementary Fig. 3E and F). Due to the relatively higher accumulation of lactate (and labelled isotopologues) in ischaemic kidneys compared with ischaemic ccRCC (Fig. 3F), this skewed the ratio of lactate $m + 3$ levels in ischaemia, thereby affecting accurate isotopic interpretation. Coincidentally, as the total lactate pool was significantly higher in ccRCC in vivo (Fig. 2B), the unlabelled ($m + 0$) lactate ratio (ccRCC vs kidney), which accounts for the majority of the total lactate pool (Supplementary Fig. 2C) was preserved between conditions (Fig. 3E and Supplementary Fig. 3E). Moreover, significant perturbation to HP isotopologues ($m + 0$, $m + 1$, $m + 3$, and $m + 6$) were

observed in ischaemic tissues due to depleted levels of this metabolite, particularly in ischaemic kidneys (Fig. 3F and Supplementary Fig. 3F and G). The depleted levels of HP isotopologues in the kidneys led to the drastic Log2FC reversal observed in ischaemic compared with control ccRCC tumours, as well as significantly altering the isotopic labelling ratios between tissues (Fig. 3E and Supplementary Fig. 3F). Interestingly, fractional enrichment (FE) analysis led to subtle differences in data interpretation (Supplementary Fig. 3G). Here, less appreciable changes were observed in HP $m + 1$ and $m + 3$ between ccRCC and kidneys in vivo compared to relative abundance levels (Supplementary Fig. 3F). Notably, unlabelled ($m + 0$) HP enrichment was proportional to the unlabelled relative abundance levels in tissues in vivo. In ischaemia, HP labelling enrichment in ccRCC vs kidneys were inconsistent with relative abundance levels, with more subtle changes observed in $m + 1$ enrichment and discordant changes observed in HP $m + 3$. Importantly, unlabelled HP enrichment in ischaemic tissues did not remain proportional to the unlabelled relative abundance levels, as often intuitively anticipated. In contrast, the HP pool (and $m + 0$) in ischaemic ccRCCs were significantly higher compared with ischaemic kidneys (Supplementary Fig. 3G). If pool sizes are unaccounted for, this could mislead inferences about reduced unlabelled and total HP pools in ccRCCs. Overall, HP labelling of $m + 1$ and $m + 3$ were key determinants in elucidating the suppressed gluconeogenic activity in our tumours in vivo. Given the contradictions in ischaemia, it is critical to utilise orthogonal approaches in isotopic tracer analysis for comprehensive biological insights in vivo, which may otherwise be masked in tissues exposed to ischaemia.

We then performed a comprehensive evaluation of the proteome and transcriptome between the in vivo and WIT conditions leveraging on the Signature Regulatory Clustering (SiRCle) model[29]. By categorising molecular features into 'biological clusters' based on the Log2FC (ccRCC vs kidney) both in vivo and in WIT conditions, we identified condition-specific changes (Supplementary Fig. 3H, Supplementary Table 2). The majority of features maintained consistent Log2FC ratios in both conditions i.e. Both_UP, Both_DOWN, and Both_no change, corroborating our findings of overall proteome and transcriptome stability in ischaemia (Fig. 3G). Over-

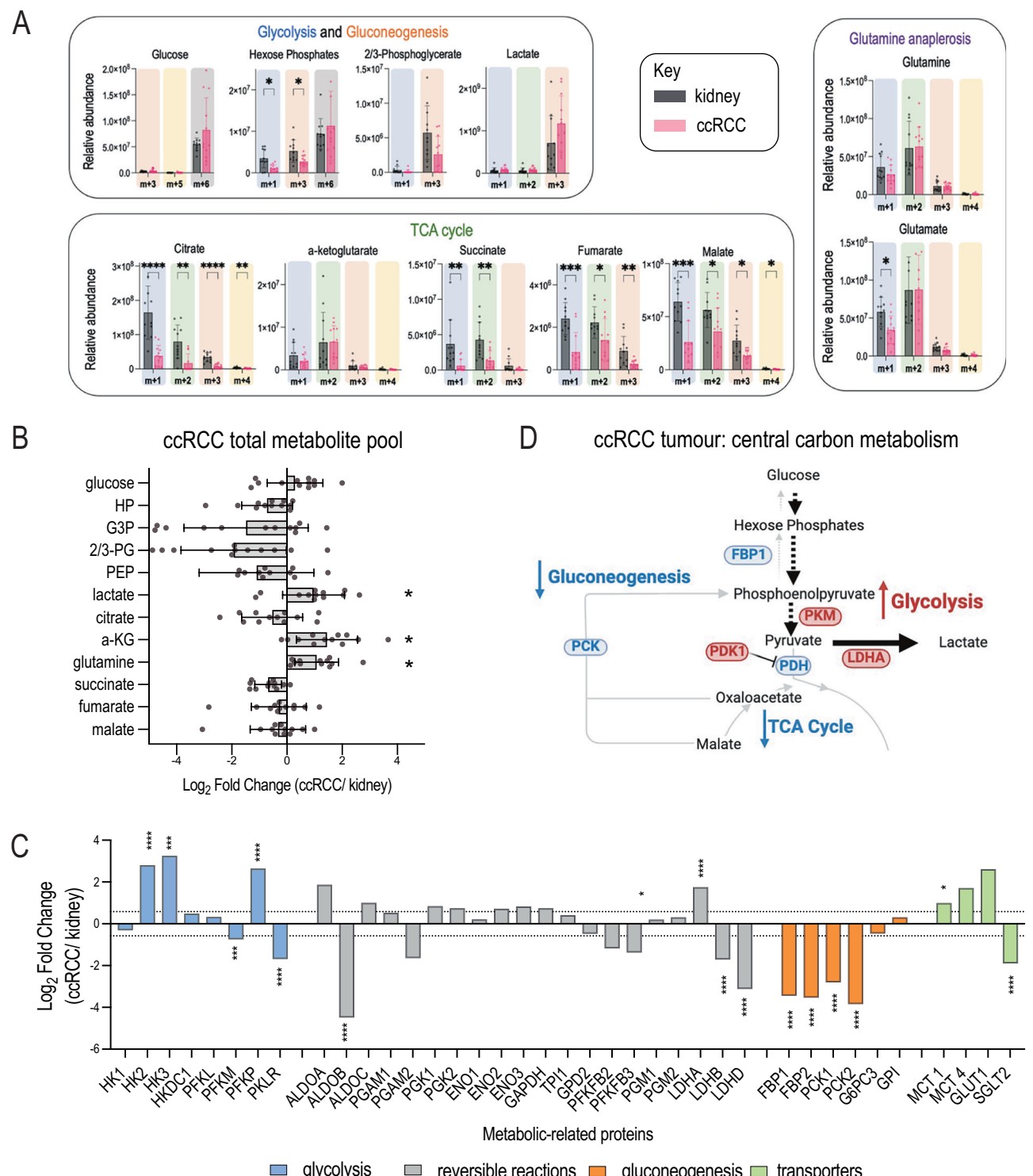

representation analyses applied to these biological clusters using the hallmarks gene set did not yield any results for the WIT_UP or WIT_DOWN clusters, likely due to the small number of features detected in these clusters (Fig. 3G). Therefore, we next focussed in on the Hallmark hypoxia gene set[30], a set of 200 genes reported to be upregulated in response to hypoxia, and mapped all detected features at the protein and RNA level (ischaemic ccRCC vs ischaemic kidneys) to this signature with biological clustering applied, to examine more granular changes (Fig. 3H and Supplementary Fig. 3I). Firstly, we observed that the majority of both protein and RNA features were common to 'both' biological clusters i.e. in vivo and

ischaemic conditions. Secondly, there was a broad distribution of Log2FC values, with elevated and reduced protein and RNA levels observed in ischaemic ccRCC tissues compared to ischaemic kidneys. These findings suggest that ischaemia does not potentiate a stronger hypoxic signature compared to in vivo tissues at the protein and RNA level (Fig. 1E). Interesting, a few metabolic enzymes such as haem-oxygenase-1 (HMOX1) and glyceraldehyde-3-phosphate (GAPDH) showed 'WIT_UP' specific clustering (Fig. 3H and Supplementary Data Fig. 3I). These enzymes are also strongly associated with the oxidative stress response[31,32]. Despite overall stability in the proteome and transcriptome, this data may suggest

**Fig. 2 | Gluconeogenesis is suppressed in ccRCC tumours in vivo.** See also Supplementary Fig. 2. **A** Labelled isotopologue profiles of indicated metabolites comparing ccRCC (pink) and kidney (black) from central carbon metabolism pathways. $n = 5$ patients, between 1 and 3 tissue samples were taken from each patient's kidney and ccRCC tumour, see Supplementary Table 1. $p = 0.015$ (Hexose Phosphates m + 1), $p = 0.023$ (Hexose Phosphates m + 3), $p = 0.00007$ (citrate m + 1), $p = 0.01$ (citrate m + 2), $p = 0.00001$ (citrate m + 3), $p = 0.005$ (citrate m + 4), $p = 0.010$ (succinate m + 1), $p = 0.006$ (succinate m + 2), $p = 0.001$ (fumarate m + 1), $p = 0.038$ (fumarate m + 2), $p = 0.009$ (fumarate m + 3), $p = 0.0004$ (malate m + 1), $p = 0.025$ (malate m + 2), $p = 0.010$ (malate m + 3), $p = 0.025$ (malate m + 4), and $p = 0.027$ (glutamate m + 1). **B** Metabolite Log2FC (ccRCC vs matched kidneys) of the total pools (labelled + unlabelled) of the indicated metabolites. $n = 5$ patients, between 1 and 3 tissue samples were taken from each patient's kidney and ccRCC tumour, see Supplementary Table 1. $p = 0.014$ (lactate), $p = 0.017$ (a-ketoglutarate), and $p = 0.013$ (glutamine). **C** Protein Log2FC (ccRCC vs kidney) for indicated proteins from the metabolic gene signature sets (MGSS) glycolysis and gluconeogenesis pathway, colour-coded by metabolic function. $p = 0.00003$ (HK2), $p = 0.007$ (HK3), $p = 0.0003$ (PFKM), $p < 0.00001$ (PFKP), $p = 0.00004$ (PKLR), $p < 0.00001$ (ALDOB), $p = 0.001$ (PFKFB3), $p < 0.00001$ (LDHA), $p < 0.00001$ (LDHB), $p < 0.00001$ (LDHC), $p < 0.00001$ (FBP1), $p < 0.00001$ (FBP2), $p < 0.00001$ (PCK1), $p < 0.00001$ (PCK2), $p = 0.032$ (MCT1), and $p = 0.00001$ (SLGT2). **D** Schematic of the dysregulated central carbon metabolism pathways in ccRCC tumours. *$p < 0.05$, **$p < 0.01$, ***$p < 0.001$, **** $<0.0001$ by two-sided Student's t-test (**A**, **C**) or two-sided paired t-test (**B**) using Benjamini-Hochberg correction. Data are mean ± SD. Source data are provided as a Source Data file. (**D**) created in BioRender. Burge, S. (2025) https://BioRender.com/n43g446.

that there are subtle degrees of ischaemia-induced perturbation with a WIT of up to 60 minutes, which should also be considered for relevant human tissue molecular studies.

### Impact of prolonged ischaemia on tissue metabolomics

Due to the highly variable WIT observed in our patient cohort and the intrinsic challenge of controlling this aspect during cancer surgery, we next investigated the effects of WIT duration on the metabolic profiling of tumours in mice. Orthotopically xenografted (*VHL*-mutant ccRCC 786-O human cells) mice underwent a $^{13}C_6$-glucose infusion for 60 minutes prior to matched tissue sampling (tumour and kidney) (Fig. 4A and Supplementary Fig. 4A). Tissues were subjected to varying durations of WIT (0, 5, 30, 60 mins) prior to metabolic quenching as previously described[13].

Notably, tissues displayed a proportional accumulation of metabolic markers of ischaemia (aside from succinate) with increasing WIT duration, with significant accumulation observed in the longer WIT groups (30–60 mins) (Fig. 4B). Unsupervised cluster analysis of the metabolome demonstrated clustering of 0 (control) and 5 mins WIT tissues and clustering of 30 and 60 mins WIT tissues for both kidney and 786-O tumours (Supplementary Fig. 4B). Furthermore, correlative analysis revealed very strong tissue-specific correlation (Rho >0.99) at 5 mins WIT compared to control conditions, which reduced with longer WIT (30–60 mins) (Supplementary Fig. 4C). This trend reduced more dramatically in the kidneys exposed to prolonged ischaemia compared with the tumour metabolome, which remained relatively stable. We also observed striking similarities with our patient data in the metabolic profiling and $^{13}C$ labelling of central carbon metabolism activity in the tumours subjected to ischaemia. Here, prolonged ischaemia also induced Log2FC reversal in HP levels of 786-O tumours exposed to long (30–60 mins) versus short (0-5 mins) WIT, whilst most metabolites also displayed differences in Log2FC amplitude between conditions (Fig. 4C). Labelled metabolite isotopologues likewise were either depleted or accumulated in ischaemic tissues relative to control conditions, and strikingly, largely exhibited a (WIT) time-dependent pattern (Fig. 4D). This was also evidenced by significant fold change differences between long and short WIT groups for labelled metabolites such as HP m + 6 and phosphoenolpyruvate (PEP) m + 3 in these 786-O tumours (Supplementary Fig. 4D). Interestingly, some metabolites such as lactate m + 3 displayed no differences between WIT groups. However, due to the proportional accumulation of labelled lactate in both tissues with increasing WIT, this preserved the fold change ratio irrespective of ischaemia duration (Fig. 4D). Overall, these results indicate that ischaemia affects metabolomics results in a time- and tissue-dependent fashion, in a comparable direction between human tumours and human xenografts in mice.

### Discussion

Conventional snapshot metabolomics studies paved the way to establishing metabolic reprogramming as a hallmark of cancer[33,34].

However, these studies have limited power to determine pathway flux[9] leading sometimes to unsupported conclusions. Human isotopic tracer studies have been used to overcome these issues by probing individual metabolic pathways (using specific nutrients), or differentiating between multiple pathway arms stemming from a metabolite (using selectively labelled nutrients)[9]. An often-overlooked aspect of the metabolomics pipeline is tissue sampling, which can occur during or after surgery. Indeed, oncological surgeries typically involve early devascularisation of the organ/tumour to reduce intraoperative bleeding risk prior to separating the organ from the surrounding tissues (resection) before the organ can be removed for histological examination[35]. It is after the time of organ removal that research tissues are conventionally sampled and therefore unintentionally subjected to variable periods of warm ischaemia. Yet, whether post-surgical tissue sampling methods with variable degrees of tissue ischaemia affect metabolomics results has not been systematically addressed.

Whilst a paucity of studies have been conducted to assess the effects of tissue ischaemia on the tumour metabolome[13], reasonable inferences can be drawn from transplant research into organ ischaemia-reperfusion injury, a phenomena whereby disruption and restoration of the blood supply to an organ results in cellular damage[36]. In metabolomics studies of human, murine, and porcine models of warm ischaemia in vivo, elevated levels of succinate, xanthine, and hypoxanthine were found to be markers of ischaemia across several tissues including kidneys[15–17]. Mechanistically, ischaemia induced several metabolic cascades including the breakdown of purine nucleotides, leading to fumarate-overflow driven reversal of succinate dehydrogenase enzyme activity, resulting in succinate accumulation[15]. During reperfusion, this accumulated succinate was shown to drive reactive oxidative species (ROS) generation and tissue damage through reverse electron transport at mitochondrial complex I[15]. Overall, these findings underscore the significant impact of environmental factors such as warm ischaemia on the metabolic responses of normal tissues.

Despite this knowledge, methodologies from RCC metabolomics studies to date, contain sparse details on how human tissues were sampled. In most cases, a lack of detail was evident, with tissues described as 'obtained after nephrectomy'[14,18,20,37,38], from tissue banks[6,39,40], or from historic cases[41], with no reference to sampling conditions or exposure to ischaemia. Our interdisciplinary approach with urologists enabled us to investigate the impact of ischaemia on tissue metabolism in ccRCC. Here, we sampled tissues in situ intraoperatively via a core biopsy needle, whilst the organ had an intact blood supply, and at the conventional sampling timepoint, that is, as soon as feasible post-surgical resection. Of note, WIT in open surgical cases remained highly variable (10-60 mins), which is likely reflective of tissues generally acquired from surgical cases.

Intraoperative sampling corroborated the aerobic glycolysis and suppressed glucose oxidation in RCC, with this metabolic phenotype

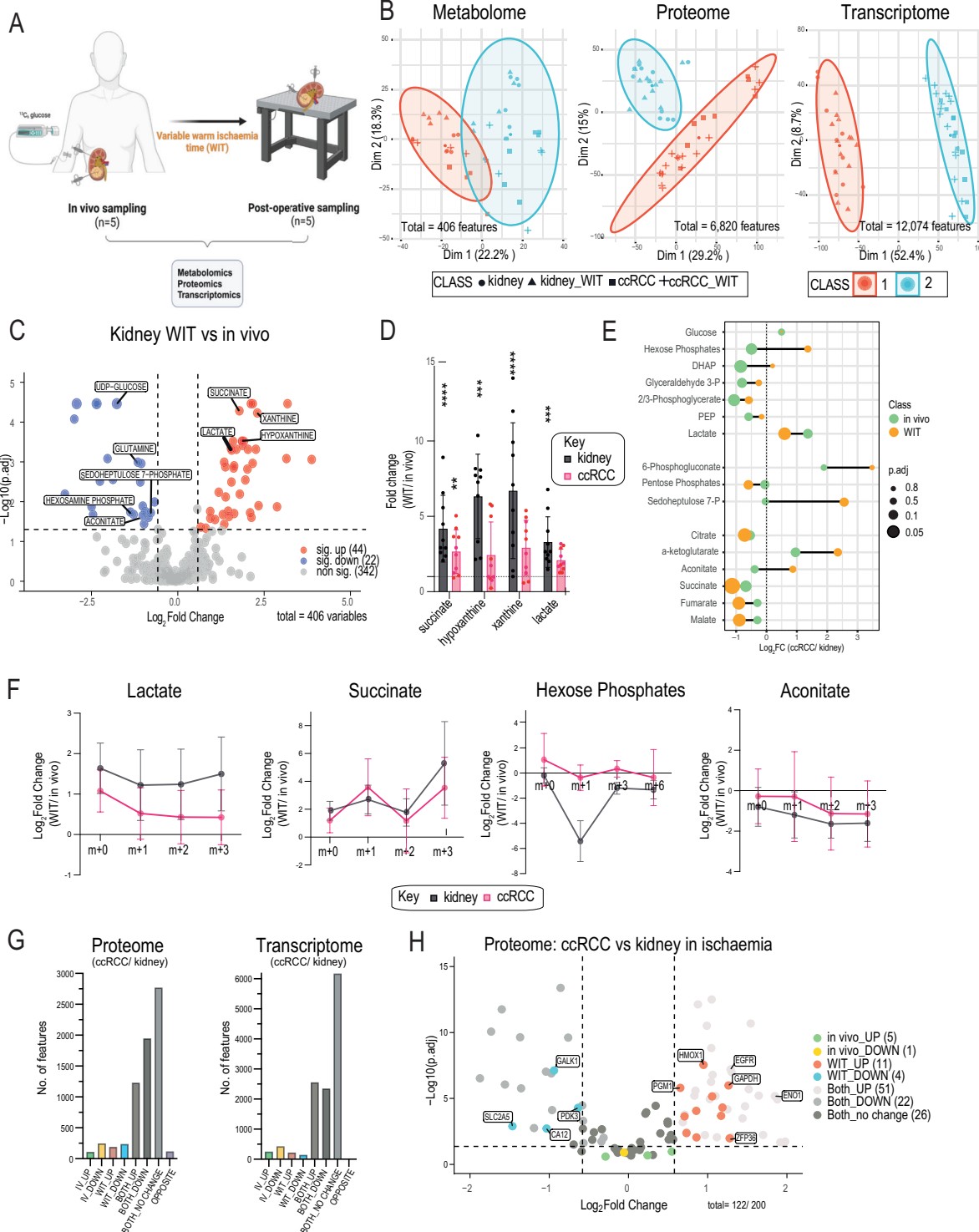

**Fig. 3 | Post-operative ischaemia perturbs the metabolic landscape of ccRCC tumours. See also Supplementary Fig. 3. A** Schematic of patient tissue sampling obtained in vivo and post-surgical resection ('Warm ischaemia time'; WIT). **B** PCA plots on indicated omics datasets with applied k-means clustering algorithm for all in vivo and WIT tissue samples. Symbol shape denotes tissue type and condition, symbol colour denotes k-means grouping. **C** Volcano plot illustrating the metabolic differences between kidney samples in in vivo and WIT conditions. Symbol colour denotes significantly up (red), significantly down (blue), and non-significant scores (grey). **D** Metabolite fold change of indicated ischaemia metabolic markers in WIT tissues compared to in vivo. $n = 5$ patients, between 1 and 3 tissue samples were taken from each patient's kidney (black) and ccRCC tumour (pink) in both conditions (i.e. in vivo and WIT), see Supplementary Table 1. Kidney: $p = 0.0005$ (succinate), $p = 0.0003$ (hypoxanthine), $p = 0.00006$ (xanthine), $p = 0.0005$ (lactate); and ccRCC

$p = 0.017$ (succinate). **E** Lollipop plot of the indicated central carbon metabolite Log2FC (ccRCC vs kidney) comparing in vivo (green) and WIT conditions (orange). Symbol size denotes p-value. **F** Tissue-specific isotopologue expressions of the Log2FC (WIT vs in vivo) for indicated central carbon metabolites. Symbol colour denotes tissue type. **G** Number of significant protein and RNA features categorised into each biological cluster based on the SiRCle model (Supplementary Data Fig. 3G). **H** Volcano plot of the Hallmark Hypoxia gene set features detected in the proteome comparing ischaemic ccRCC vs ischaemic kidneys, colour-coded based on the biological cluster classification (Supplementary Data Fig. 3G). $*p < 0.05$, $**p < 0.01$, $***p < 0.001$, $**** < 0.0001$ by two-sided Student's t-test using Benjamini-Hochberg correction (**C**, **D**, **E**, and **H**). Data are mean ± SD. Source data are provided as a Source Data file. (**A**) created in BioRender. Burge, S. (2025) https://BioRender.com/n43g446.

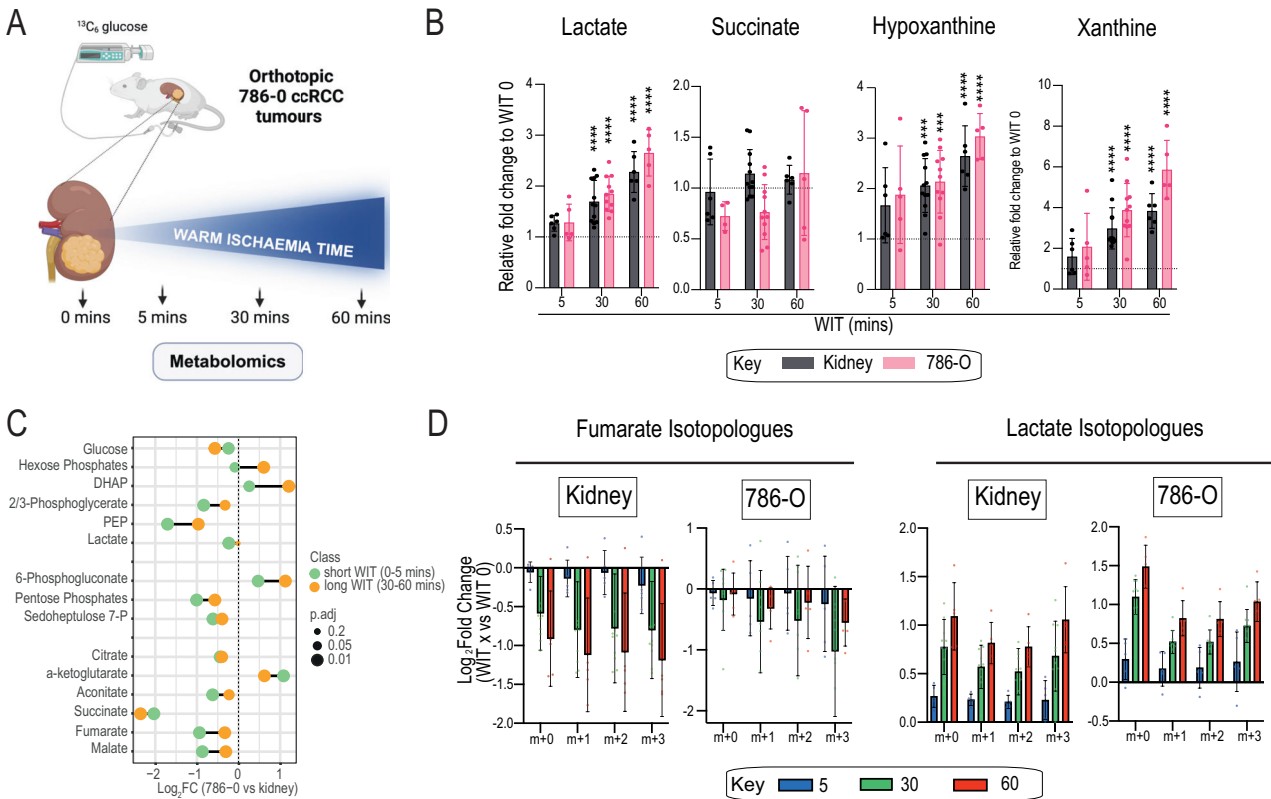

**Fig. 4 | Impact of prolonged ischaemia in 786-O tumour metabolism in vivo. See also Supplementary Fig. 4. A** Schematic of mouse kidney and orthotopic 786-O tumour sampling at indicated warm ischaemia timepoints. $n = 5$-12 mice at each WIT timepoint. **B** Metabolite fold change of ischaemic metabolites markers at the indicated WIT relative to no ischaemia. For 786-O tumours (pink), mice WIT 0 mins $n = 11$; WIT 5 mins $n = 5$; WIT 30 mins $n = 11$; and WIT 60 mins $n = 5$. For mouse kidneys (black), mice WIT 0 mins $n = 12$; WIT 5 mins $n = 6$; WIT 30 mins $n = 12$; and WIT 60 mins $n = 6$. Kidney: lactate $p = 0.00002$ (WIT 30), $p < 0.00001$ (WIT 60); hypoxanthine $p = 0.0001$ (WIT 30), $p < 0.00001$ (WIT 60); and xanthine $p < 0.00001$(WIT 30), $p < 0.00001$ (WIT 60). 786-O: lactate $p = <0.00001$ (WIT 30), $p < 0.00001$ (WIT 60); hypoxanthine $p = 0.0001$ (WIT 30), $p = <0.00001$ (WIT 60);

and xanthine $p < 0.00001$(WIT 30), $p < 0.00001$ (WIT 60). **C** Lollipop plot of the indicated metabolite Log2FC (786-O vs kidney) comparing short (0-5 mins; green) and long (30-60 mins; orange) WIT groups. Symbol size denotes p-value. **D** Metabolite isotopologue expressions of the Log2FC (WIT time point vs no ischaemia) for individual tissues. For 786-O tumours, mice WIT 0 mins $n = 11$; WIT 5 mins $n = 5$; WIT 30 mins $n = 11$; and WIT 60 mins $n = 5$. For mouse kidneys, mice WIT 0 mins $n = 12$; WIT 5 mins $n = 6$; WIT 30 mins $n = 12$; and WIT 60 mins $n = 6$. Bars are colour-coded based on WIT time. *$p < 0.05$, **$p < 0.01$, ***$p < 0.001$, **** $<0.0001$ by one-way ANOVA (**B**) and two-sided Student's t-test using Benjamini-Hochberg correction (C). Data are mean ± SD. Source data are provided as a Source Data file. **A** created in BioRender. Burge, S. (2025) https://BioRender.com/n43g446.

preserved in ccRCC tissues sampled in vivo and in ischaemic conditions. Importantly, through our multiomics and orthogonal approach to labelled metabolomics, we evidenced the suppression of gluconeogenic activity in human ccRCC tumours in vivo. Our findings confirm previous interrogations of The Cancer Genome Atlas (TCGA) ccRCC RNA-sequencing data, alongside near-uniform depletion of FBP1 in >200 human ccRCC tumours, which revealed depletion of key gluconeogenic enzymes[39]. These findings now established in vivo may be of clinical interest given ectopic FBP1 expression in *VHL*-deficient ccRCC cells 'reverted' reprogrammed tumour metabolism, directly inhibited nuclear HIF function, and inhibited tumoural growth in xenografted mice[39]. Interestingly, we also observed a heterogeneous distribution of argininosuccinate even if ASS1 and ASL were overall suppressed in these tumours. Given that re-expression of ASS1 levels with corresponding increase of argininosuccinate was a crucial event for metastatic ccRCC cells to invade in vitro and metastasise in vivo in a mouse model[42], these results may indicate that in some primary tumours, cells may re-express ASS1, becoming primed to metastasise. Arginine or aspartate tracer studies in patients with ccRCC are warranted to confirm such findings before exploring clinically actionable targets.

In the post-resection samples, we found that ischaemia had a significant impact on the metabolic landscape of RCC, with the proteome and transcriptome remaining broadly stable. Given the

variability in time-scale for metabolic fluxes i.e. glycolytic time-scale of seconds to minutes[43], compared to other biological processes such as protein and RNA synthesis and/or modification, alongside the knowledge that the metabolome captures the final downstream interaction between the genome and the environment, this may explain why changes in the metabolome, in response to cues such as ischaemia, are amplified compared to changes upstream at the protein or RNA level. Still, the observation that enzymes typically associated with oxidative stress, such as HMOX1, are moderately upregulated may suggest that even within these timeframes, WIT can influence the molecular landscape of ccRCC. The possible oxidative stress experience during WIT may also explain the changes in aconitate, as the enzyme aconitase is an established redox-sensitive enzyme[44].

Here, we demonstrated that ischaemia altered labelled metabolite isotopologues in a tissue-specific manner, which could dramatically reverse labelling ratios, including in metabolites (i.e., labelled HP) that were a key determinant in inferring suppressed gluconeogenesis in vivo. This critical technicality may underlie some of the discrepancies reported in human ccRCC metabolomics studies to date, including contradictory findings of significantly accumulated versus significantly depleted succinate levels in two of the largest patient studies to date[6,37]. As succinate is a recognised marker of tissue ischaemia, these inconsistencies may be reflective of study methods. More recently, purine nucleotide metabolism has

been implicated in various mammalian tumours[45,46], given ours and previous studies have shown that purine nucleotide breakdown products are significantly elevated in ischaemia compared to in vivo, this may broaden the implications of our findings further. One could argue that snapshot metabolite levels cannot differentiate between altered metabolite consumption or production, however we also demonstrated the significant impact on $^{13}C$ -labelling of central carbon metabolism activity in these tumours. We subsequently expanded upon these findings in our ccRCC mouse model, where we determined that it was the duration of WIT, rather than exposure to ischaemia, that was the key factor to the dysregulation observed. Here, we showed that the tissue metabolome remained stable with <5 mins of WIT exposure, whereas prolonged durations of ischaemia (30-60 mins) largely recapitulated the changes to the metabolic profile and $^{13}C$ -labelling observed in our patient tumours (Supplementary Fig. 5).

Interestingly, Johnston et al. reported that the labelling profiles of human tumour samples, and in xenografted mice, subjected to WIT of >30 mins remained stable compared to those sampled immediately[13]. In this study, comparative analysis was conducted in a range of extracranial solid paediatric tumours (8 distinct tumours with up to 5 subtypes). As tumours can exhibit distinctly labelled metabolic profiles[14], this may account for our difference in findings. Furthermore, in our study, tumours were implanted orthotopically, whereas tumours were subcutaneously xenografted in their study. Studies in lung, glioma, and prostate cancer have shown that subcutaneously xenografted tumours in mice compared to orthotopic tumours had lower vascular perfusion and higher hypoxic burdens[47–49]. Although unreported in these studies, this hypoxia-inducing microenvironment potentially stimulated a compensatory molecular response in tumours implanted subcutaneously, which may precondition them to further hypoxic insult. This theoretically could have contributed to the labelled metabolome stability observed in ischaemic tumours in their study[13]. Finally, given the well characterised pseudohypoxic phenotype of *VHL*-mutant ccRCC tumours due to aberrant HIF stabilisation[50,51] and the resultant HIF-induced metabolic sequelae, we postulate that these types of tumours may be also somewhat pre-conditioned or metabolically more 'robust' to ischaemic insults than benign kidney tissues. Hence, ischaemia-induced metabolic changes may also go undetected, as observed in the accumulated lactate in these ccRCC tumours irrespective of ischaemic exposure. We postulate this may be due to possible convergence on established tumour metabolic (e.g. hypoxic) phenotypes, thereby 'camouflaging' this specific ischaemic perturbation.

In summary, we have provided evidence that prolonged tissue ischaemia, a by-product of conventional tissue sampling from surgeries, significantly perturbs the metabolic characterisation and labelling of ccRCC. Furthermore, by matching patient tissue samples taken intraoperatively and post-surgical resection, we provided the ideal control group to demonstrate the impact of ischaemia on the tumour metabolome. Consequently, this study has critical implications on future human tissue sampling methods within our research community, particularly in costly and resource-intense isotopic tracer studies conducted in patients. These findings strongly advocate for standardised tissue sampling methods that minimise ischaemic exposure, such as the in situ tissue sampling method we describe, which has been implemented into the clinical trial setting[52], in order to facilitate precise molecular characterisation of tissues for clinically translatable research.

## Limitations of the study

We intentionally recruited patients with ccRCC undergoing open surgery (compared to laparoscopic or robotic surgery) because research tissues sampled at the typical time point (post-resection) would generally have a shorter WIT, due to the nature of organ removal once the specimen is separated from surrounding tissues, and for better vascular access and control in the event of intraoperative bleeding from intraoperative sampling.

The generalisability of our findings is likely to be restricted due to sample size and similar tumour staging and grading (i.e. advanced, high grade ccRCC tumours), with aggressive ccRCC tumours reported to exhibit additional metabolic rewiring compared to early stage tumours[4,6,22].

We have not sought to elucidate the mechanisms of ischaemia-induced metabolic perturbation in our cohort of tumours. During ischaemia, cells undergo a series of complex pathological processes involving anaerobic metabolism, inflammation, dysfunction, and once a threshold is exceeded, irreversible injury and/or cell degradation[53]. Whilst this may be academically interesting, it would most certainly be challenging to elucidate and of likely limited benefit in our goal to characterise the tumour metabolome. Instead, our focus should be on optimising tissue sampling methods by minimising exposure to ischaemia (<5 mins) to accurately profile tumour metabolism.

Lastly, as this study was performed in surgical patients undergoing a general anaesthetic (who are also fasted), we must consider that the physiological responses to anaesthesia and surgery may affect the interpretation of the tumour molecular biology. Indeed, we and others have demonstrated that the tissue metabolome is particularly sensitive to external cues such as ischaemia. Whilst little is known about the direct effects on human tissue metabolism, the literature is replete with studies exploring the metabolic sequelae of anaesthesia in human plasma samples and preclinical rodent models[54–58]. To address this important topic, particularly as surgery has been an ideal setting for human isotopic tracer studies to date, our future work includes conducting matched isotopic tracer studies in a cohort of patients with RCC undergoing radiologically-guided renal biopsies (non-fasted, awake under a local anaesthetic) prior to surgery, with our established intraoperative tissue sampling methods to further assess the impact of these external factors on the tumour metabolome.

## Methods

All research performed in this study complies with all relevant ethical regulations. Patient study ethics were approved by the UK Human Research Authority, East of England- Cambridge Central Research Ethics Committee (Ref: 19/EE/0161). All animal procedures were approved by the UK Home Office (Project Licence No. PFCB122AA) and the University of Cambridge Animal Welfare and Ethical Review Body.

### Human participants

Adults were identified and enrolled into this study from uro-oncology multidisciplinary team meetings (MDT) and/ or renal oncology clinics by the clinical team at Addenbrooke's Hospital, CUHFT following informed consent from all participants. See Table 1 for patient demographics. Eligible patients were further discussed with the operating surgeon and the anaesthetic study lead (V.P) to ensure suitability of patients for labelled-glucose infusions and intraoperative sampling. Eligibility criteria included adults (aged 18 years and over) diagnosed with a 'renal mass' undergoing a nephrectomy as part of their clinical management at CUHFT.

### Animal models

All experiments were conducted in strict accordance with the UK Animals (Scientific Procedures) Act 1986 by personnel with the appropriate personal licence. Eleven healthy male NSG mice (Charles River Laboratories, UK) were xenografted at between 8-13 weeks of age. All mice were housed in specific-pathogen-free animal facilities with *ad libitum* access to food via a hopper (Diet: R105-25, Product code: U8407G10R, SAFE) and water. Dark/light cycle was 12 hours:12 hours, ambient temperature parameter 20-24 degrees celcius, with the humidity set between 45-65%. Prior to the described studies, mice were

monitored regularly and determined to be healthy by the veterinary staff. A maximal tumour burden of 12 mm was permitted under the animal licence approved by the UK Home Office and this parameter was not exceeded in this study.

## Cell lines

The 786-0 human male ccRCC cell line (American Type Culture Collection; CRL-1932) was cultured in physiological Plasmax media (Ximbio, UK). The cells were engineered to stably express luciferase as previously described[59]. Mycoplasma negativity was confirmed by the MycoAlert™ Mycoplasma Detection Kit (Lonza, LT07-318). The 786-O cells were not authenticated after purchase.

## Manufacture of $^{13}C_6$- glucose for human studies

Clinical grade (cGMP) $^{13}C_6$ D-glucose was obtained from Merck, Germany and formulated to UK Good Manufacturing Practice (GMP) standards for human intravenous administration at Stockport Pharmaceuticals, UK. Product was specially formulated as a 20% glucose w/v injection solution in 10 ml glass ampoules to enable precise quantities to be administered whilst minimising wastage. The formulation process was also overseen by the CUHFT Pharmacy Department (A.C.).

## Patient infusions with $^{13}C_6$- glucose

Dosing regimen for intraoperative $^{13}C_6$-glucose infusions was based on previous studies[10–12,14]. On the day of surgery, an initial bolus of 8 g over 10 mins was administered by the anaesthetist to the patient via a peripheral cannula immediately after anaesthetic induction. This was followed by a continuous infusion of 4-8 g/h until surgical resection of the tumour-bearing kidney. Serial blood samples were taken from an arterial line inserted by the anaesthetist at baseline (pre-bolus) and at approximately 30 minutes intervals until surgical resection, for the purposes of monitoring blood glucose and measuring $^{13}C$ enrichment of metabolites. Depending on blood glucose concentration levels (target 9-11 mmol/L), as measured by point of care testing, the continuous infusion rate was titrated accordingly. Serum samples were stored on wet ice until the procedure was complete, after which they were centrifuged (1000 g, 10 min, 4 °C) with the supernatant stored at −80 °C until analysis.

## Patient tissue sampling

Preoperative radiological imaging was reviewed by the operating team prior to surgery to plan for intraoperative tissue sampling. Immediately prior to renal artery ligation, the operating urologist biopsied the tumour and adjacent kidney using separate coaxial 14 G biopsy needles (Achieve #A1415, UK) to avoid cross-contamination. The coaxial device permitted several biopsies to be taken through a single tissue puncture site minimising tissue trauma and bleeding. Multiregional samples were acquired at random by angulating the biopsy device within the tumour and kidney. A sling was used to identify and isolate the renal artery vessel prior to biopsy in case timely ligation was required for excessive bleeding from biopsy sites. In addition, haemostatic agents such as Surgicel® (Johnson & Johnson, USA) were used to tamponade bleeding from biopsy sites during multi-regional sampling. We aimed to obtain 3 biopsies from both the tumour and adjacent kidney, however this was terminated if there were any clinical concerns or complications intraoperatively. Once the specimen was resected, it was immediately delivered to research personnel within the operating theatre for (post-operative) tissue sampling. Tissues were re-sampled as described above. The duration of warm ischaemia time (WIT), that is, the time taken from renal arterial ligation to post-operative sampling was recorded for each case. At the time of tissue sample collection, all tissues were snap frozen in liquid nitrogen and stored at −80 °C until analysis. For the various molecular analyses (metabolomic, proteomics, and DNA/ RNA sequencing), individual frozen tissue samples

were split on dry ice into 3 equal parts to facilitate complementary analyses on the same tissue sample. Where samples were smaller than expected, tissues were prioritised for metabolomics, followed by proteomics or DNA/RNA extraction. All planned molecular analyses were performed on at least one tumour and kidney sample from an individual patient. Tumour histology was confirmed by a Consultant Uro-pathologist (A.W).

## Mouse xenografts and infusions

On the day of surgery, 786-O-luciferase ccRCC cells ($1 \times 10^6$ per mouse kidney injection) was reconstituted in a 1:3 ratio of Matrigel (BD Biosciences): culture media (total volume of 50 μl/ injection) and stored on wet ice until the procedure. Mice were anaesthetised and positioned in the right lateral position. An oblique 10 mm left flank incision was made in the skin and muscle overlying the left kidney and the left kidney exteriorised. The 786-O cell suspension was then injected (Hamilton 250 μl syringes with a blunt needle, UK) under the renal capsule and the kidney returned to the abdomen and layers closed with absorbable sutures (Safil®, Braun, Germany) and surgical clips (Braintree Scientific, USA). All surgeries were performed under an operating microscope (Leica #M641, UK). Tumour growth was monitored regularly using a bioluminescence imager (IVIS® spectrum, PerkinElmer, USA) and Living Image software (PerkinElmer, USA) after subcutaneous injections with 100 μl D-Luciferin (Merck, USA) reconstituted to 30 mg/ml in PBS. Labelled $^{13}C_6$-glucose (Cambridge Isotopes Laboratory, USA) infusions were performed in mice when the normalised tumour flux (p/s) was at least 4-fold compared to baseline readings, which correlated with ~8-12 mm tumour growth. Labelled glucose mouse infusions was conducted as previously described[60]. In brief, $^{13}C_6$-glucose was prepared at a dose of 2.48 g/kg mouse weight dissolved in 750 μl saline and administered as a 125 μl bolus over 1 min followed by a continuous infusion at 2.5 μl/min for 60 mins. After anaesthetic induction, mice were cannulated via the lateral tail vein and infusion immediately started using a syringe pump (World Precision Instruments, Aladdin #6-220, UK). This procedure was performed simultaneously in 4-6 mice to minimise inter-experimental variation. Mice were euthanised at the end of the infusion and tissues immediately harvested. Tissue samples (both kidney and 786-O tumour) were divided into 4 equal pieces, one piece was immediately flash frozen in liquid nitrogen and the other pieces left at room temperature in a petri dish for 5, 30, or 60 mins prior to freezing, as previously described[13].

## Liquid-chromatography mass spectrometry

Tissue samples were homogenised (6000 rpm, 2 ×30 secs, Precellys®24 tissue homogeniser, Bertin Instruments, France) in an extraction buffer (25 μl/mg tissue) containing 50% methanol, 30% acetonitrile, 20% ultrapure water, and 5 μM valine-d8 (CK isotopes, UK) as an internal standard. In total, 43 human tissue samples from 5 patients (22 ccRCC, 21 kidney) were analysed. Sample controls were the respective patient kidney tissues and there were 1-3 biological replicates, please see Supplementary Table 1. Samples were incubated on dry ice for 20 mins, mixed (Thermomixer, Eppendorf, Germany, 500 g, 15 mins, 4 °C) and then centrifuged (16,000 g, 20 mins, 4 °C) with the supernatants stored at −80 °C until analysis. Plasma samples were thawed on wet ice, vortexed, and 40 μl of each sample added to 360 μl extraction buffer, as described. Samples were vortexed, mixed (6000 g, 15 mins, 4 °C) and centrifuged (16,000 g, 20 mins, 4 °C) with the supernatants storing at −80 °C. Separate pooled samples for tissue and plasma were compiled and interspersed at regular intervals within the sample sequence for quality control (QC) analysis. Samples were analysed on a Q Exactive Hydrid Quadrupole-Orbitrap Mass spectrometer (Thermo Scientific, USA) coupled to a Dionex Ultimate 3000 UHPLC (Dionex, USA). HILIC chromatographic separation of metabolites was achieved using a Millipore Sequant ZIC-pHILIC analytical column (5 μm, 2.1 × 150 mm, Merck, USA) equipped with a 2.1 × 20 mm

guard column (both 5 mm particle size) with a binary solvent system. Solvent A contained 20 mM ammonium carbonate, 0.05% ammonium hydroxide; and Solvent B was acetonitrile. The column oven and autosampler tray were held at 40 °C and 4 °C, respectively. The chromatographic gradient flow rate was run at 0.200 mL/min as follows: 0–2 min: 80% B; 2-17 min: linear gradient from 80% B to 20% B; 17-17.1 min: linear gradient from 20% B to 80% B; 17.1-22.5 min: hold at 80% B. The mass spectrometer was operated in full-scan, polarity-switching mode. The spray voltage was set to +4.5 kV/−3.5 kV, the heated capillary maintained at 320 °C, and auxiliary gas heater maintained at 280 °C. The sheath gas flow was set to 25 units, the auxiliary gas flow was set to 15 units, and the sweep gas flow was set to 0 unit. HRMS data acquisition was performed in a range of $m/z = 70–900$, with the resolution set at 70,000, the automatic gain control (AGC) target at $1 × 10^6$, and the maximum injection time (Max IT) at 120 ms. Acquired spectra were analysed using Tracefinder 5.0 software (Thermo Fisher, USA) with referencing to an internal library of compounds and integrated peak areas used for semi-quantification of detected metabolites. Metabolite identities were confirmed using two parameters: the precursor ion $m/z$ was matched ±5 ppm of theoretical mass predicted by the chemical formula; and secondly, the retention time of the metabolite was within 5% of the retention time of a purified standard run with the same chromatographic method. Samples were randomised to avoid bias due to machine drift and analysed in a blinded manner. Pooled samples were interspersed at regular intervals within the sample sequence as quality control. For $^{13}C$-labelling analysis, the theoretical masses of $^{13}C$ isotopes were calculated and added to a library of predicted isotopes. These masses were then searched with a 5 ppm tolerance and integrated if the peak apex showed <1% difference in retention time from the $[U^{-12}C]$ monoisotopic mass in the same chromatogram. Hexose phosphates encompassed glucose 6-phosphate and fructose 6-phosphate, as detected by this platform. Samples were randomised to avoid bias due to machine drift and analysed in a blinded manner. Natural isotope abundances were corrected for using the AccuCor algorithm (https://github.com/lparsons/accucor). Additional control samples in the form of separate unlabelled patient samples of kidney and tumour tissues were run simultaneously to check natural isotopic abundance corrections. Raw data (ion intensity) was normalised to the total ion count of each sample to give the relative abundance (arbitrary units) of that metabolite. All measurements were taken from distinct samples.

## Proteomics

Tissue samples (3-8 mg) were homogenised (6000 rpm, 2 ×30 secs, Precellys®24 tissue homogeniser) in 50 μl lysis buffer containing 6 M Guanidine Hydrochloride, 100 mM Trisaminomethane pH 8.5, Chloroacetamide 1 mg/ml, and Tris(2-carboxyethyl)phosphine hydrochloride 1.5 mg/ml. In total, 37 human tissue samples from 5 patients (17 ccRCC, 20 kidney) were analysed. Sample controls were the respective patient kidney tissues and there were 1-3 biological replicates, please see Supplementary Table 1. Samples were then centrifuged (16,000 g, 2 mins, 4 °C) and heated to 95 °C for 5 mins. Protein quantification was performed (Bradford Assay, Abcam, UK; and Infinite® M200 Pro, Tecan, Switzerland) before being stored at −80 °C until analysis. On the day before analysis, samples were heated to 97 °C for 5 mins, then Lys-C pre-digest (Fujifilm WakoPure Chemical Corporation, Japan) was added (1 μL/sample) with a 3-hour incubation at 37 °C. 200 μl Mass Spec grade water and 1 μg trypsin (Promega) were then added to samples before overnight incubation at 37 °C with hard shaking. On the day of analysis, samples were acidified with 1% TFA and centrifuged (18,000 g, 10 mins, RT) before applying samples to a double-layer Empore C18 Extraction Disk (3 M) prepared with methanol. Membrane was washed twice with 0.1% TFA and protein was eluted with elution buffer (50% ACN, 0.05% TFA), dried using a CentriVap Concentrator (Labconco) and resuspended in 15 μl 0.1% TFA. Protein

concentration was determined by absorption at 280 nm on a Nanodrop 1000, with 2 μg of de-salted peptides loaded onto a 50 cm emitter packed with 1.9 μm ReproSil-Pur 200 C18-AQ (Dr Maisch, Germany) using an RSLC-nano uHPLC systems connected to a Orbitrap Fusion Lumos mass spectrometer (both ThermoFisher Scientific, USA).

Raw files were analysed, quantified, and statistical analysis performed using Spectronaut 15 pipeline (Biognosis, Switzerland) using directDIA against the Uniprot Human database (uniprot-proteome-UP000005640-human-isoforms.fasta 15-March-2021) with the default settings. Specifically: fixed modification Carbamidomethylation (C), variable modification Acetylation (protein N-terminus), Oxidation (M). Digest Type: Specific Trypsin/P cleavage with maximum 2 missed cleavages, mass accuracy (Dynamic), Peptide Length:7-52, False Discovery Rate (FDR) Peptide FDR: 0.01, Protein Group FDR: 0.01, PSM FDR: 0.01.

## RNA sequencing

Tissue RNA was extracted (AllPrep DNA/RNA Mini Kit, Qiagen, Germany) and quantified using the 4150 Tapestation system (Agilent, cat.no. G2992AA). Library preparation was performed using SMART-seq v4 low-input mRNA (Takara Bioscience, cat.no. 634893). After obtaining cDNA, the Nextera XT workflow (Illumina, cat.no. FC-131-1096) was used to perform the adapter ligation using the tagmentation protocol. The libraries for each sample were then quantified and pooled together using both the Qubit 2.0 (Invitrogen) and qPCR system (Aria MX real time PCR system, Agilent). The KAPA library quantification kit (Roche) was used for qPCR. Once pooled, the final library was quantified again by qPCR before analysis using the NextSeq 500 (Illumina) with the high output 75 cycle kits (Ilumina). QC of the reads was performed using FastQC (v0.11.4, https://www.bioinformatics.babraham.ac.uk/projects/fastqc/). Reads were trimmed using TrimGalore (v0.5.0, https://www.bioinformatics.babraham.ac.uk/projects/trim_galore/). Base calling after sequencing was done using the software provided by Illumina (bcl2fastq). Reads were mapped to the genome using STARmapper (v2.7.1[61]) and Ensembl Homo Sapiens GRCh38 (release 105) reference genome using the annotated transcripts from the ensembl Homo_Sapiens.GRCh38.105.gtf file. The number of reads that map to genomic features was calculated using HTSeq v0.6.1[62]. Reads with a poor mapping quality (<10), mapping to multiple loci, or to overlapping gene regions were discarded to avoid ambiguity and false positives. Batch correction was performed using the COMBAT_seq correction method from the SVA package[63]. Normalisation and differential gene expression analysis was performed using the edgeR package (v3.26.5[64]).

## DNA gene panel analysis

Tissue RNA was extracted (AllPrep DNA/RNA Mini Kit, Qiagen) and quantified using the DNA dsDNA BR Assay Kit assay kit (Invitrogen) and the Qubit Fluorometer 3.0 (Invitrogen). Libraries were prepared using a custom gene panel including the *VHL* gene (Total Size: 1455082 base pairs) from TWIST Biosciences (Cat No.101281: Twist Library Preparation Kit). Samples were analysed using the Nextseq 2000 (Illumina). Sequencing data was analysed using the in-house Cancer Molecular Diagnostics Laboratory (CRUK, Cambridge, UK) GCP compliant pipeline (version 0.41) which utilised the following main algorithms: bwa-mem (hg38 alt contig aware alignment), samtools (PCR Duplicates removal), GATK (Base Quality Score, Recalibration, InDel Realignment), GATK's Haplotype Caller (Germline Variant Calling), GATK's MuTect2 (Somatic Variant Calling), GATK (CNV Calling), Delly (Non-CNV SV Calling), and Annovar (Variant Annotation).

## Statistics

Data was tested for normal distribution using the Shapiro-Wilks test prior to statistical analysis. If data significantly deviated from normality ($p < 0.01$), data was $\log_2$-transformed and tested again for normality.

For parametric data, student t-tests were used for non-paired data and paired t-test for paired data. For multiple comparisons, one-way analysis of variance (ANOVA) test was used. p-values were adjusted using the Benjamini-Hochberg method to reduce the false discovery rate (FDR) and the (adjusted) p-value < 0.05 was used as the statistically significant threshold. For correlative analysis, Spearman's method and the LOESS (locally weighted smoothing) regression line was used. Hierarchical cluster analysis was performed using euclidean distance with the clustering Ward's linkage algorithm. K-means clustering analysis was performed using the Hartigan and Wong algorithm on the first 2 principal components with the number of clusters set to $n = 2$, the random starting assignment specified at 25 and the maximum number of iterations set at 10[65]. Gene set enrichment analyses was performed using the GSEA software (v4.2.2, Broad Institute[66,67]). Permutations were set at 1000 and gene set databases used included the Molecular Signatures Database (MSigDB) Gene set 'h.all.v7.5.1.symbols.gmt [Hallmarks]'[30] as well as a manually curated published metabolic gene signature set (MGS)[25]. FDR value of 0.25 for GSEA was used as the statistically significant threshold. The ranking metric used for GSEA was the Log2fold change in descending order for all features and the normalised enrichment score (NES) was used to compare results across gene sets. The SiRCle model[29] was used to categorise protein and RNA features into biological clusters based on a significant fold change ratio threshold ≥1.5 or ≤ 0.67 with an adjusted p-value < 0.05. All analyses were performed using either R Studio (v 4.1.2), GraphPad Prism v9, and Metaboanalyst v5.0, a web-based analytical pipeline for metabolomics data[68,69]. Adobe Illustrator v28.7.1 and BioRender software (https://app.biorender.com/) was used to create paper schematics.

### Reporting summary

Further information on research design is available in the Nature Portfolio Reporting Summary linked to this article.

## Data availability

The mass spectrometry proteomics data have been deposited to the ProteomeXchange Consortium via the PRIDE partner repository with the dataset identifier PXD054588. The RNA sequencing data has been deposited to the Gene Expression Omnibus (GEO) data repository with the accession no: GSE274774. The metabolomics data has been deposited to the Metabolomics Workbench repository[70] with the study ID: ST004207 (https://doi.org/10.21228/M8VC26). Source data are provided with this paper.

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

## Acknowledgements
We thank all the patients who participated in this study as well as our NHS colleagues; urologists, anaesthetists, radiologists, scientists, and pathologists. Marco Sciacovelli provided assistance in culturing the cell lines for xenografting. Alyson Speed, Dr. Nikola Dolezalova, and Jackie

Higgins provided guidance and training in the animal-related studies. Prof. Ralph J DeBerardinis (RJB), Dr Akash Kaushik, Dr Brandon Faubert, and the laboratory of RJB provided guidance and training in the human isotopic tracer studies. Cambridge Genomics Services (Lead: Dr Julien Bauer, Dept. of Pathology, University of Cambridge, UK) and Cancer Molecular Diagnostics Laboratory (CRUK, Cambridge, UK) for assistance with the RNA sequencing and DNA gene panel analysis and bioinformatics analysis, respectively. Dr. Anita Chhabra (Lead Clinical Trials Pharmacist, Addenbrooke's Hospital, CUHFT) and Stockport Pharmaceuticals for assistance with labelled glucose formulation for patient administration. The study was supported by the Wellcome Trust (RG92001), The Urology Foundation Scholar Award (G100328), and the Fulbright Commission to C.Y. The Mark Foundation for Cancer Research [RG95043], the Cancer Research UK Cambridge Centre [C9685/A25177 and CTRQQR-2021\100012] and NIHR Cambridge Biomedical Research Centre (NIHR203312) to G.D.S and S.A. Cancer Research UK Cambridge Centre [C9685/A25177] and NIHR Cambridge Biomedical Research Centre (BRC-1215-20014) to A.Y.W. Medical Research Council (MC_UU_12022/7), Deutsche Forschungsgemeinschaft (DFG, German Research Foundation) under Germany's Excellence Strategy (EXC 2030 – 390661388) and Alexander-von-Humboldt professorship awarded to C.F. Marie Curie (722605), and CRUK Cambridge Centre [C51061/A27453] to C.S. Sigrid Jusélius Foundation (N/A) to S.V. The views expressed are those of the authors and not necessarily those of the NIHR or the Department of Health and Social Care.

## Author contributions

C.Y., G.D.S., and C.F. conceived of the project. C.Y., G.D.S., and V.P. managed the clinical protocol, consented and infused patients. C.Y., G.D.S., J.N.A., A.C.P.R., T.J.M., and A.Y.W. collected tissue samples from the infused patients. A.Y.W. reported the pathology. M.Y. acquired metabolomics data. C.Y., C.S., G.D.S., M.Y., and C.F. processed, analysed, and interpreted data from the labelled glucose infusions. A.V.K. acquired the proteomics data. C.Y., C.S., A.V.K., G.D.S., and C.F analysed and interpreted the proteomics data. C.Y., C.S., G.D.S., and C.F. analysed and interpreted the RNA sequencing data. S.A acquired, processed, and analysed the DNA gene expression data. S.V. provided the 786-O-luciferase cells for xenografting. S.V. and K.S.P. provided guidance and training for animal-related studies. C.Y. performed the mouse infusions. C.Y. performed the statistical analysis. C.Y., G.D.S., and C.F. wrote the manuscript. All authors contributed to reviewing and editing the final manuscript.

## Funding

## Competing interests

G.D.S has received educational grants from Pfizer, AstraZeneca and Intuitive Surgical; consultancy fees from Pfizer, MSD, EUSA Pharma and CMR Surgical; Travel expenses from MSD and Pfizer; Speaker fees from Pfizer; Clinical lead (urology) National Kidney Cancer Audit and Topic Advisor for the NICE kidney cancer guideline. The remaining authors declare no competing interests.
