## [Transparent Peer Review file · Nature Communications]

Tumour sampling conditions perturb the metabolic landscape of clear cell renal cell carcinoma

Corresponding Author: Professor Christian Frezza

This manuscript has been previously reviewed at another journal that is not operating a transparent peer review scheme. The manuscript was considered suitable for publication without further review at Nature Communications.

Version 0:

Reviewer comments:

Reviewer #4

(Remarks to the Author)

This is an important study that seeks to characterize the impact of devascularization and tissue ischemia on molecular and metabolic readouts in the context of in vivo tissue sampling of ccRCC. The authors take an orthogonal approach, obtaining data from both patients and a mouse model. This is a transfer submission that has been revised based on prior reviewer comments.

Overall, I think the authors have addressed several of the comments satisfactorily. However, I recommend the following minor modifications prior to publication:

1. As stated by prior reviewers, fractional enrichment data is critical for understanding the relative contributions of the pathways being discussed. The Hexose Phosphate fractional enrichment data should be moved from the supplement to the main figures. It would be valuable for the authors to directly articulate for the reader what is being extensively discussed in the reviews – i.e. gluconeogenic activity is not the primary pathway utilized in human kidneys or ccRCC tumors but likely has biological consequences and is not captured from in vivo studies that don't control for ischemia. The isotopologue relative abundance data and the fractional enrichment data support these points. I would simplify and improve readability of Lines 167-173.

2. Related to point 6 of the Response to Reviewers, I agree that the authors have clearly described the changes that occur during ischemia and the importance of minimizing ischemic time. However, the abstract and manuscript are still currently written with too strong of wording that implies prior and future studies that fail to control for ischemia are broadly unreliable, which is not supported by the data. The abstract and discussion still do not convey two important points of the paper:

A) Many metabolic features and relative ratios are preserved despite ischemia. For example, in Extended Fig. 4C, the correlation coefficient was ~ 0.98 for 786-O tumor cells after one hour of WIT and ~ 0.925 for kidney tissue after one hour of WIT. Furthermore, most ratios (Fig. 3E, Fig. 4C) remain directionally consistent regardless of sampling conditions.

Additionally, the current study supports and validates many findings from prior studies where ischemia was not controlled. For example, ccRCC demonstrate suppressed TCA labeling in vivo regardless of sampling condition. The entire metabolic landscape is not changed by ischemia; just some of it. This nuance needs to be articulated in the abstract (eg 1 sentence) and the discussion (eg 1 paragraph).

B) In both the patient and PDX studies of this paper, normal kidney was much more metabolically impacted by ischemia than ccRCC tumors. The abstract should acknowledge this important finding.

3. Fig. 3C: the authors show a volcano plot of Kidney WIT vs in vivo. It would be valuable to show the parallel volcano plot of ccRCC WIT vs in vivo next to this. Data from both volcano plots should be available in supplementary tables.

4. Fig. 3E: change dot sizes so that 0.5, 0.1, and 0.05 are appreciably different by eye.

5. Ext. Fig. 3D needs clarification. Title says Lactate (WIT); color coding suggests it's in vivo. Perhaps they should just show both groups as they did for hexose 6-phosphate below.

I commend the authors for a well-done study, which I think is an important addition to the field.
